# Noise-Aware Few-Shot Learning through Bi-directional Multi-View Prompt Alignment

## Abstract

Vision-language models offer strong few-shot capability through prompt tuning but remain vulnerable to noisy labels, which can corrupt prompts and degrade cross-modal alignment. Existing approaches struggle because they often lack the ability to model fine-grained semantic cues and to adaptively separate clean from noisy signals. To address these challenges, we propose NA-MVP, a framework for **N**oise-**A**ware few-shot learning through bi-directional **M**ulti-**V**iew **P**rompt alignment. NA-MVP introduces three key innovations: multi-view prompts with unbalanced optimal transport alignment that enable fine-grained patch-to-prompt matching while suppressing unreliable regions and reinforcing clean correspondences; a bi-directional prompt design that jointly models clean-oriented and noise-aware semantics to disentangle useful signals from corrupted ones; and adaptive sample refinement with optimal transport that employs a learnable threshold to correct mislabeled samples while preserving reliable data. Experiments on both synthetic and real-world noisy datasets demonstrate that NA-MVP consistently outperforms state-of-the-art baselines, highlighting its effectiveness for robust few-shot learning under noisy supervision.

## 1 Introduction

Vision-language models (VLMs), such as CLIP (Radford et al., 2021), have significantly advanced multimodal understanding by embedding visual and textual information into a shared semantic space. This powerful alignment facilitates a wide range of downstream tasks through prompt learning, which optimizes a set of learnable textual embeddings to guide predictions while keeping the backbone parameters frozen (Zhou et al., 2022b;a). Prompt learning is particularly attractive in few-shot and resource-limited settings due to its parameter efficiency, modularity, and rapid adaptability. Despite these advantages, VLMs remain vulnerable in real-world applications where label noise is prevalent, which can severely mislead prompt optimization and degrade model robustness.

Recent studies show that prompt learning can be resilient to label noise, motivating its integration with learning with noisy label methods (Wu et al., 2023). Existing approaches include negative learning (Sun et al., 2022; Wei et al., 2024) and noisy label selection (Guo & Gu, 2024; Pan et al., 2025), but they still face key limitations, as illustrated in Figure 1. First, prompt expressiveness is constrained, since most methods employ only one or two learnable prompts (a positive and a negative pair) (Wu et al., 2023; Wei et al., 2024), enforcing a single-view alignment that cannot capture diverse and fine-grained cues. Intuitively, a comprehensive understanding of an image often depends on observing it from multiple viewpoints. Second, assigning an explicit negative label to each image imposes a rigid supervision signal tied to a pre-specified counter class, which can bias prompt learning under asymmetric or class-dependent noise and under-represent the diversity of corrupted patterns. Third, denoising is typically coarse, with labels refined using confidence-based rules or pseudo labels that ignore inter-sample dependencies and the global semantic structure, allowing errors to propagate rather than be selectively suppressed.

To address the challenges of limited prompt expressiveness and inaccurate label refinement, we propose NA-MVP, a framework tailored for few-shot learning under noisy labels. NA-MVP couples multi-view, fine-grained patch-to-prompt alignment with Unbalanced Optimal Transport (UOT) to align local CLIP features with prompts, emphasize semantically relevant regions, and suppress noisy content. It then adopts a bi-directional prompt architecture in which each class uses clean-oriented

Figure 1: Limitations of existing prompt learning approaches under noisy labels. **Single-view reliance**: Limited prompts miss diverse visual patterns. **Explicit negatives**: Fixed negatives impose rigid supervision. **Fixed threshold**: Coarse denoising lets noise propagate.

and noise-aware prompts optimized jointly while treating non-target classes as implicit negatives, enriching semantic coverage and maintaining prompt diversity. Finally, a prompt-guided selective label refinement module employs a learnable threshold derived from bi-directional alignment signals to identify likely mislabeled samples, and only those are corrected via classical Optimal Transport, preserving reliable labels and stabilizing training. Because the multi-view and bi-directional components are tightly coupled aspects of our prompt design, we present them together in the method section. Our main contributions are summarized as follows:

- We propose multi-view prompt learning with Unbalanced Optimal Transport under noisy labels, achieving fine-grained patch-to-prompt alignment that downweights unreliable regions and strengthens clean matches.

- We introduce a bi-directional prompt design that jointly learns clean-oriented and noise-aware prompts to provide complementary semantics, maintain diversity, and separate clean from corrupted signals.

- We develop prompt-guided selective label refinement with classical Optimal Transport, using a learnable threshold from bi-directional alignment to refine only likely mislabeled samples while preserving clean labels.

- We validate NA-MVP on multiple benchmarks and noise settings, showing consistent gains and robustness under noisy supervision.

## 2 RELATED WORK

**Learning with noisy labels.** LNL presents a significant challenge in training models that generalize well without overfitting to noisy labels. Existing approaches include robust loss functions (Zhang & Sabuncu, 2018; Wang et al., 2019; Ma et al., 2020; Wei et al., 2023), loss correction (Chang et al., 2017; Arazo et al., 2019; Xia et al., 2019), robust noise regularization (Wei et al., 2020; 2021; Iscen et al., 2022; Ko et al., 2023), and sample selection (Kim et al., 2021; Karim et al., 2022; Feng et al., 2023; Huang et al., 2023; Li et al., 2023; Xu et al., 2023; Wei et al., 2024). Sample selection methods often rely on the small-loss criterion, with a risk of mistakenly discarding clean hard samples and retaining noisy ones. Label correction strategies, like MLC (Zheng et al., 2021) and SELC (Lu & He, 2022), aim to correct noisy annotations by generating pseudo-labels from model predictions. However, these methods typically process each sample independently, overlooking the relationships between data points, which can lead to suboptimal corrections. To better exploit global structure, recent works (Xia et al., 2022; Feng et al., 2023) use OT to align noisy and clean label distributions, while CSOT (Chang et al., 2023) further incorporates intra-distribution relationships for improved pseudo-labeling. Recently, prompt learning has shown promise in noisy settings (Wu et al., 2023; Wei et al., 2024). For instance, NLPrompt (Pan et al., 2025) uses a prompt-based OT to distinguish clean and noisy samples by aligning vision-language features. Building on these ideas, we propose multi-view prompt learning with optimal transport for noisy label learning. By leveraging pre-trained vision-language models, NA-MVP enhances sample selection and feature alignment, enabling robust training under label noise.

**Prompt Learning in Vision-Language Models.** Prompt learning, initially developed in natural language processing, has become a key technique in VLMs (Jia et al., 2021; Radford et al., 2021; Yu et al., 2022). Early models like CLIP used hand-crafted prompts, but recent methods focus on learning continuous prompts. CoOp (Zhou et al., 2022b) introduced learnable prompts in the continuous embedding space, with CoCoOp (Zhou et al., 2022a) further adapting these prompts for each image, improving generalization to unseen classes. This shift has inspired numerous studies (Shu et al., 2022; Derakhshani et al., 2023; Khattak et al., 2023a;b; Liu et al., 2023; Roy & Etemad, 2023; Zhu et al., 2023) to enhance prompt learning. It has been successfully applied to a variety of tasks, such as robust learning (Wu et al., 2023; Pan et al., 2025; Wei et al., 2024), semantic segmentation (Lüddecke & Ecker, 2022; Rao et al., 2022), and federated learning (Guo et al., 2023; Yang et al., 2023; Li et al., 2024). However, learning a single prompt (Zhou et al., 2022b) overlooks the diversity of visual representations. Recent works (Chen et al., 2022; Lu et al., 2022; Sun et al., 2022) have explored multiple prompts to address this. For example, CLIPN (Wang et al., 2023) uses positive and negation prompts for out-of-distribution detection. EMPL (Sun et al., 2022) generates multiple prompt embeddings through an energy-based distribution, improving open-vocabulary generalization. In contrast, our method constructs learnable multi-view prompts in a bi-directional manner, combining clean-oriented and noise-sensitive representations to jointly model reliable semantics and potential label noise patterns. This design enables the model to capture richer class characteristics and improves its robustness under noisy supervision.

**Optimal Transport.** OT provides a principled way to compare probability distributions by finding the most efficient mapping between them at a given cost. It defines the Wasserstein distance (Peyré et al., 2019) and has been widely adopted in machine learning and computer vision. However, the high computational complexity of OT was a bottleneck until Cuturi introduced entropic regularization, enabling efficient computation via the Sinkhorn algorithm (Distances, 2013). To improve flexibility, UOT (Lombardi & Maitre, 2015; Chizat et al., 2018) was introduced, replacing the strict mass conservation constraint in classical OT with soft penalization terms (Frogner et al., 2015; Lahn et al., 2023). These advances have enabled OT to support a wide range of applications, including semi-supervised learning (Taherkhani et al., 2020; Lai et al., 2022; Wang et al., 2022), object detection (Ge et al., 2021; Yang et al., 2021; De Plaen et al., 2023), generative models (Balaji et al., 2020; Daniels et al., 2021; Choi et al., 2023), domain adaptation (Chang et al., 2022; Wang et al., 2024), learning with noisy labels (Feng et al., 2023; Chang et al., 2023) and others. Building on these advances, our method adopts UOT with relaxed mass constraints to align local image features with multi-view prompts, allowing the model to focus on reliable features while suppressing noise. Meanwhile, classical OT with strict mass preservation is used to refine noisy labels by aligning global image features with class-level prompts, ensuring reliable label correction. This design leverages the complementary strengths of both OT variants for robust learning under label noise.

## 3 METHODOLOGY

Our method consists of two key components: Bi-directional multi-view prompts for noise-aware alignment, where objects are observed from multiple perspectives. Selective noisy label refinement with OT, where the refinement is guided by prompts, as shown in Figure 2. We present multi-view and bi-directional as a unified prompt design, since the two are tightly coupled in how they provide complementary semantics and generate alignment signals for refinement.

**Problem Definition.** Let $\mathcal{D}_{\text{noisy}} = \{(x_i, y_i)\}_{i=1}^{D}$ denote the noisy training set with images $x_i$ and labels $y_i$ from $C$ classes. However, whether the given label is accurate or not is unknown. We classify the correctly labeled instances as clean, and the mislabeled ones as noisy. The goal of LNL is to train a model that maintains high test accuracy while minimizing the influence of label noise.

### 3.1 BI-DIRECTIONAL MULTI-VIEW PROMPTS FOR NOISE-AWARE ALIGNMENT

To improve robustness under noisy supervision, we propose a bi-directional multi-view prompt learning strategy that integrates multiple clean-oriented and noise-aware prompts for each class. Unlike negative learning (Wei et al., 2024), which relies on explicitly labeled negative classes, our method treats all unlabeled classes as implicit negatives, avoiding the need for additional annotations. Clean-oriented prompts focus on capturing class-relevant semantics, while noise-aware

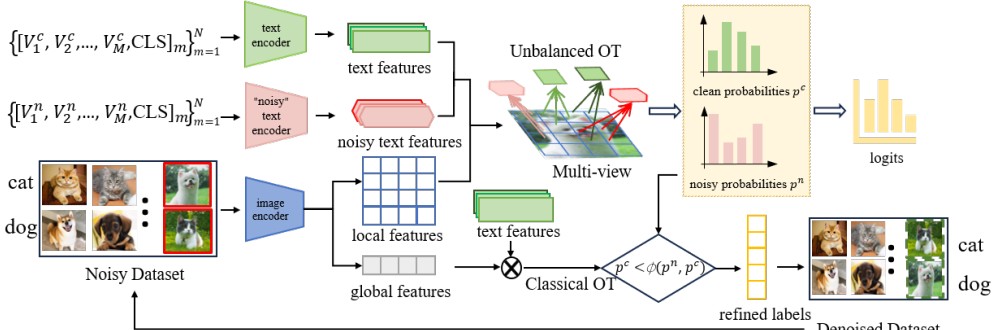

Figure 2: Overview of the proposed NA-MVP framework. Our framework consists of two key modules: (1) Bi-directional multi-view prompts for noise-aware alignment, where clean and noise-aware prompts are encoded separately and aligned with local image features via UOT to obtain clean and noisy probabilities for noise-aware feature alignment and prediction; (2) Selective noisy label refinement, where samples are identified as noisy based on a threshold computed from clean and noisy probabilities, and refined using classical OT on global features to correct labels. The two modules work together to produce a denoised dataset and robust predictions under noisy supervision.

prompts function as adaptive filters to identify and suppress misleading or noisy signals. These two types of prompts are jointly optimized to facilitate noise-aware representation learning and enhance the model's discrimination between clean and corrupted samples. This bi-directional prompt mechanism offers complementary perspectives on the data and serves as the foundation for downstream label refinement and denoising.

**Bi-directional Multi-view Prompt Construction and Feature Encoding.** For each class k, we construct two sets of learnable prompts: clean-oriented prompts $\{\text{Prompt}_{m,k}^c\}_{m=1}^N$ and noise-aware prompts $\{\text{Prompt}_{m,k}^n\}_{m=1}^N$, where $N$ denotes the number of prompts for each class. Each prompt consists of M context tokens followed by a class-specific token:

$$\text{Prompt}_{m,k}^c = [V_1^c, V_2^c, \ldots, V_M^c, \text{CLS}_k] \tag{1}$$

$$\text{Prompt}_{m,k}^n = [V_1^n, V_2^n, \ldots, V_M^n, \text{CLS}_k] \tag{2}$$

where $V_i^c$ and $V_i^n$ are learnable word embeddings, and $\text{CLS}_k$ denotes the class-specific token. Clean and noisy prompts are encoded by two separate text encoders $\psi(\cdot)$ and $\psi^n(\cdot)$ to generate prompt feature sets $\boldsymbol{G}_k^c \in \mathbb{R}^{N \times d}$ and $\boldsymbol{G}_k^n \in \mathbb{R}^{N \times d}$, where d is the feature dimension. Given an input image $\boldsymbol{x}_i$, the image encoder extracts a global feature $\boldsymbol{f}_i \in \mathbb{R}^d$ and a local feature map $\boldsymbol{F}_i = \{f_l\}_{l=1}^L \in \mathbb{R}^{L \times d}$, where $L = H \times W$, with $H$ and $W$ denoting the height and width of the feature map.

**Fine-grained Noise-aware Alignment.** To align local image features with multi-view prompts under noisy conditions, we employ UOT for noise-aware multi-view alignment. Unlike classical OT, UOT relaxes the strict mass conservation constraint, allowing partial alignment and reducing overfitting to noise. This enables more selective matching between informative visual features and semantic prompts, preserving prompt diversity and improving alignment robustness.

Specifically, we treat $\boldsymbol{F}_i$ and $\boldsymbol{G}_k$ as discrete distributions and compute the cost matrix $\boldsymbol{C}_k = 1 - \boldsymbol{F}_i^\top \boldsymbol{G}_k \in \mathbb{R}^{L \times N}$ by using the cosine similarity. The UOT problem is then formulated as:

$$d_{\text{UOT}}(k) = \min_{T \in \Pi(\mu,\nu)} \langle \boldsymbol{C}_k, T \rangle \tag{3}$$

$$\Pi(\mu,\nu) = \left\{ T \in \mathbb{R}_+^{L \times N} \mid T\mathbb{1}_N \le \mu, T^\top \mathbb{1}_L = \nu \right\} \tag{4}$$

where $\mu \in \mathbb{R}^L$ and $\nu \in \mathbb{R}^N$ represent marginal probability vectors satisfying $\|\mu\|_1 \ge \|\nu\|_1 = \theta$, with $\theta \in [0, 1]$ controlling the degree of partial matching. The relaxed constraints in UOT allow for partial alignment between image features and prompt embeddings, rather than enforcing full correspondence. This is particularly helpful under label noise, where forcing all features to align

may introduce irrelevant or incorrect information. To efficiently solve the UOT problem, we apply the Sinkhorn distance with entropic regularization:

$$d_{\text{UOT}}(k) = \min_{T \in \Pi(\mu, \nu)} \langle \boldsymbol{C}_k, T \rangle + \epsilon \langle T, \log T \rangle, \tag{5}$$

This formulation can be interpreted as a Kullback-Leibler (KL) projection (Benamou et al., 2015), where the feasible set is defined by two convex constraints:

$$d_{\text{UOT}}(k) = \min_{T \in \Pi(\mu, \nu)} \epsilon \, \text{KL}(T \mid e^{-\boldsymbol{C}_k/\epsilon}) \tag{6}$$

$$\mathcal{C}_1 \stackrel{\text{def.}}{=} \left\{ T \in \mathbb{R}_+^{L \times N} \mid T \mathbb{1}_N \leq \mu \right\}, \quad \mathcal{C}_2 \stackrel{\text{def.}}{=} \left\{ T \in \mathbb{R}_+^{L \times N} \mid T^\top \mathbb{1}_L = \nu \right\} \tag{7}$$

To solve Eq. 6, we use a fast implementation of Dykstra's algorithm, which scales the iterative KL projection between $\mathcal{C}_1$ and $\mathcal{C}_2$ using matrix-vector multiplications. The optimization proceeds iteratively, initializing $Q = \exp(-\boldsymbol{C}_k/\epsilon)$ and $\nu^{(0)} = \mathbb{1}_N$. After a few iterations, the transport plan $T^*$ is computed as:

$$T^* = \text{diag}(\mu^{(t)}) Q \text{diag}(\nu^{(t)}) \tag{8}$$

where $t$ is the iteration number. In each iteration, the scaling factors are updated as $\mu^{(t)} = \min(\mathbb{1}_L/Q_\mu \nu^{(t-1)}, \mathbb{1}_L)$ and $\nu^{(t)} = \mathbb{1}_N/Q_\nu^T \mu^{(t)}$ with $Q_\mu = Q/\text{diag}(\mu)\mathbb{1}_{L \times N}$ and $Q_\nu = Q^T/\text{diag}(\nu)\mathbb{1}_{L \times N}$. Once the $T^*$ is obtained, we use it to compute the alignment UOT distance and optimize learnable vectors in the bi-directional multi-view prompts $\{\text{Prompt}_m^{c/n}\}_{m=1}^N$. Overall, by relaxing the strict mass conservation constraint, UOT enables partial and adaptive alignment between local features and multi-view prompts. This noise-aware alignment allows the model to attend to semantically reliable features while down-weighting noisy or irrelevant features, thereby preserving prompt diversity and enhancing robustness under label noise.

**Image-Text Bi-directional Prompt Loss.** We further stabilize this process with an auxiliary Image-Text Bi-directional Prompt (ITBP) Loss. The design of ITBP follows the bi-directional contrastive loss introduced in CLIPN (Wang et al., 2023), but its role in our framework is fundamentally different. While CLIPN leverages this loss for out-of-distribution detection, ITBP in our setting is employed to explicitly separate clean and noisy semantics. Concretely, it encourages image features to align more closely with clean prompts while simultaneously pushing them away from corresponding noisy prompts and unrelated negatives. This adaptation strengthens the effectiveness of the bi-directional multi-view prompt design under noisy label supervision.

## 3.2 PROMPT-GUIDED SELECTIVE LABEL REFINEMENT WITH OT

**Adaptive Noise Identification via Bi-directional Prompt Alignment.** To adaptively identify noisy labels, we measure the alignment between the local image feature $\boldsymbol{F}_i$ and the clean prompt $\boldsymbol{G}_k^c$ as well as the noise-aware prompt $\boldsymbol{G}_k^n$, yielding similarities $s_{i,k}^c$ and $s_{i,k}^n$; we then derive an adaptive threshold $\phi_{i,k}$ by applying a softmax to these similarities:

$$\phi_{i,k} = \frac{\exp(s_{i,k}^n/\tau)}{\exp(s_{i,k}^c/\tau) + \exp(s_{i,k}^n/\tau)}, \tag{9}$$

where $\tau$ is a temperature parameter controlling the sharpness of the threshold. Then, we classify sample $\boldsymbol{x}_i$ as clean if its similarity to the clean prompts exceeds this threshold:

$$\mathcal{D}_{\text{clean}} = \left\{ (x_i, y_i) \mid s_{i,k}^c > \phi_{i,k}, k = y_i \right\}. \tag{10}$$

Samples that do not meet this condition are considered noisy. Rather than discarding these noisy samples, we refine their labels using a pseudo-labeling strategy based on classical OT. In this way, the bi-directional prompt framework not only identifies noisy samples but also re-integrates them with corrected supervision. The clean and noisy prompts are jointly optimized with the model parameters, enabling continuous refinement and better adaptation to the evolving feature space.

**Label Refinement via Classical Optimal Transport.** Classical OT is a powerful tool for aligning distributions, and it has been effectively used for generating pseudo-labels by matching samples to class distributions while preserving the global structure of the sample distribution through equality constraints. Here, we adopt classical OT to refine noisy labels in the context of noisy label learning.

Given a noisy training set $\mathcal{D}_{\text{noisy}} = \{(x_i, y_i)\}_{i=1}^D$, we extract global image features $\boldsymbol{F} \in \mathbb{R}^{D \times d}$, and the prompt features $\boldsymbol{G} \in \mathbb{R}^{C \times d}$ using a pre-trained vision-language model. Next, we compute the similarity matrix between the image and prompt features, $\boldsymbol{G} \cdot \boldsymbol{F}^\top$. We use the negative logarithm of this matrix as the cost matrix. To ensure proper alignment, we impose uniform marginal distributions across both the samples and the classes. The OT problem is then formulated as:

$$d_{\text{OT}}(\mu, \nu) = \min_{\boldsymbol{T} \in \Pi(\boldsymbol{\mu}, \boldsymbol{\nu})} \langle -\log(\boldsymbol{G} \cdot \boldsymbol{F}^\top), \boldsymbol{T} \rangle \tag{11}$$

$$\Pi(\mu, \nu) = \left\{ \mathbf{T} \in \mathbb{R}_+^{C \times D} \mid \mathbf{T}\mathbb{1}_D = \mu, \ \mathbf{T}^\top \mathbb{1}_C = \nu \right\} \tag{12}$$

where $\mathbb{1}_C$ is the vector of ones with length $C$, representing the total probability mass of the noisy label distribution, and $\mathbb{1}_D$ is the vector of ones with length $D$, representing the total probability mass of the sample distribution. These constraints ensure that the total probability mass is conserved across both the samples and the labels. Once the optimal transport plan $T^*$ is computed, the pseudo-label for each image $x_i$ is obtained by selecting the class with the highest transport mass:

$$\tilde{y}_i = \arg \max_j T_{ij}^* \tag{13}$$

This process generates refined labels by using the transport plan $T^*$ to assign the most probable class for each image. To further improve reliability, we integrate the adaptive threshold $\phi_{i,k}$ defined earlier to identify potentially mislabeled samples. Only those samples whose similarity to clean prompts falls below the threshold are considered for refinement, ensuring that clean samples remain unaltered while noisy instances are corrected:

$$\mathcal{D}_{\text{refinement}} = \left\{ (x_i, \tilde{y}_i) \mid s_{i,k}^c < \phi_{i,k}, k = y_i \right\}. \tag{14}$$

The selective mechanism, built on the bi-directional multi-view prompt learning framework, enables the model to isolate and correct corrupted labels effectively. This not only improves the label quality but also enhances the robustness of the model under noisy supervision. Ultimately, the denoised training set is constructed by combining reliable clean samples with the refined noisy ones:

$$\mathcal{D}_{\text{denoised}} = \mathcal{D}_{\text{clean}} \cup \mathcal{D}_{\text{refinement}}. \tag{15}$$

By training on $\mathcal{D}_{\text{denoised}}$, the model benefits from both trustworthy clean supervision and corrected noisy labels, leading to more stable convergence and improved generalization performance.

## 3.3 TRAINING DETAILS

**Training Schedule.** To improve robustness, we delay the label refinement process and only start modifying labels after $T_{\text{sup}}$ epochs. Details of the full training procedure are provided in Appendix A. In the early phase, the model is trained on the noisy dataset with the Generalized Cross-Entropy (GCE) loss (Zhang & Sabuncu, 2018) combined with the ITBP loss:

$$\mathcal{L}_{\text{sup}} = \mathcal{L}_{\text{gce}} + \lambda_{\text{i}} \cdot \mathcal{L}_{\text{itbp}}, \tag{16}$$

where $\lambda_{\text{i}}$ controls the strength of auxiliary supervision. Once the refinement process is activated, noisy samples identified by our prompt-guided mechanism are selectively corrected, and training continues on the updated dataset with GCE loss. This delayed refinement allows the model to first acquire stable representations before adapting to cleaner supervision. A detailed sensitivity study on the effect of $\lambda_{\text{i}}$ is reported in the Appendix C.4.

**Inference.** During inference, both clean and noise-aware prompt alignments are incorporated into the prediction. Specifically, we measure the alignment of the local image feature $\boldsymbol{F}_i$ with the clean-oriented prompt feature $\boldsymbol{G}_k^c$ and the noise-aware prompt feature $\boldsymbol{G}_k^n$. The similarities between the image feature and these prompts are computed using the UOT-based distance, as defined in Eq. 3, such that:

$$s_{i,k}^c = 1 - d_{\text{UOT}}(\boldsymbol{F}_i, \boldsymbol{G}_k^c), \tag{17}$$

$$s_{i,k}^n = 1 - d_{\text{UOT}}(\boldsymbol{F}_i, \boldsymbol{G}_k^n), \tag{18}$$

where higher values of $s_{i,k}^c$ and $s_{i,k}^n$ indicate stronger semantic similarity with the clean and noise-aware prompts, respectively. The final probability of assigning label $k$ to image $x_i$ is defined as:

$$p(y = k \mid x_i) = (1 - p_{ik}^n) \cdot p_{ik}^c \tag{19}$$

$$p_{ik}^c = \frac{\exp(s_{i,k}^c / \tau)}{\sum_{j=1}^C \exp(s_{i,j}^c / \tau)}, \quad p_{ik}^n = \frac{\exp(s_{i,k}^n / \tau)}{\exp(s_{i,k}^c / \tau) + \exp(s_{i,k}^n / \tau)} \tag{20}$$

Here, $p_{ik}^c$ represents the probability derived from the alignment with the clean prompt, while $p_{ik}^n$ reflects the probability from the alignment with the noise-aware prompt. This bi-directional multi-view strategy encourages the model to prioritize clean samples while adaptively down-weighting potential noisy ones, leading to more stable and accurate learning under noisy supervision.

# 4 EXPERIMENTS

## 4.1 EXPERIMENTAL SETTINGS

**Datasets.** To evaluate the robustness of our method under label noise, we conduct experiments on five widely-used synthetic noisy datasets: Caltech101 (Fei-Fei et al., 2004), DTD (Cimpoi et al., 2014), Flowers102 (Nilsback & Zisserman, 2008), OxfordPets (Parkhi et al., 2012), and UCF101 (Peng et al., 2018). Following (Pan et al., 2025), we utilize two types of label noise: symmetric noise (Sym) and asymmetric noise (Asym). For symmetric noise, labels in the training set are randomly flipped to other classes with equal probability; for asymmetric noise, labels are flipped to semantically similar neighboring classes. In addition, we also evaluate on a real-world noisy dataset, Food101N (Lee et al., 2018), which inherently contains label noise without requiring synthetic corruption. A detailed introduction of each dataset can be found in the Appendix B.1.

**Implementation Details.** We follow the experimental protocol of CoOp (Zhou et al., 2022b) to ensure fair comparison. The model is optimized using SGD with an initial learning rate of 0.002, a momentum of 0.9 and a weight decay of $5 \times 10^{-4}$. Unless otherwise specified, ResNet-50 is used as the image encoder and a Transformer with 63M parameters as the text encoder. For each dataset, we adopt a 16-shot training setup and evaluate on the full clean test set. The default number of training epochs is set to 50. We employ 16 shared context tokens appended with the class token. All reported results are averaged over three runs with different random seeds, and the best accuracy is highlighted in bold. All experiments are conducted on a single NVIDIA GeForce RTX 4090 GPU. More implementation details can be found in the Appendix B.2.

## 4.2 PERFORMANCE COMPARISON

We compare our method with strong baselines, including CoOp (Zhou et al., 2022b), CoOp+GCE (Wu et al., 2023), JoAPR (Guo & Gu, 2024) and NLPrompt (Pan et al., 2025). Among them, CoOp serves as a standard prompt learning baseline, while CoOp+GCE, JoAPR and NLPrompt are specifically designed to address label noise in the few-shot prompt learning scenario. These comparisons allow us to comprehensively assess the effectiveness and robustness of our method in both synthetic and real-world noise settings. As shown in Table 1, our proposed NA-MVP consistently achieves the best overall performance across all five datasets. Furthermore, in scenarios with high levels of noise, our method exhibits a substantial performance advantage, highlighting its robustness in severely corrupted environments. These results demonstrate that NA-MVP effectively mitigates the impact of label noise and generalizes well under challenging settings.

We further evaluate our method on the real-world noisy dataset Food101N. Table 2 presents the test accuracy under different few-shot settings. Across all scenarios, NA-MVP consistently outperforms NLPrompt, achieving the best overall performance. Notably, as the

Table 2: Test accuracy on Food101N.

| Method | 4-shot | 8-shot | 16-shot | 32-shot |
|---|---|---|---|---|
| NLPrompt | 70.57 | 73.93 | 76.46 | 76.87 |
| NA-MVP | **76.10** | **76.27** | **76.90** | **77.03** |

number of shots decreases, NLPrompt's performance drops significantly, while NA-MVP remains stable and robust. These results highlight the effectiveness of NA-MVP in few-shot learning with real-world noise, underscoring its practicality for deployment.

## 4.3 ABLATION STUDIES AND MORE ANALYSIS

We conduct a comprehensive ablation study to quantify the contributions of each core component in our proposed NA-MVP framework. Specifically, we investigate the impact of (1) bi-directional multi-view prompt learning, (2) UOT for local feature alignment, and (3) label refinement with OT and $\phi_{i,k}$, as shown in Table 3.

Table 1: Comparison of methods under symmetric and asymmetric noise on five datasets.(%)

| Dataset | Method | Noise rate: Sym | | | | | | Noise rate: Asym | |
|---|---|---|---|---|---|---|---|---|---|
| | | 0.125 | 0.25 | 0.375 | 0.5 | 0.625 | 0.75 | 0.25 | 0.5 |
| Caltech101 | CoOp | 86.43 | 81.03 | 76.73 | 70.90 | 61.33 | 46.90 | 75.23 | 49.43 |
| | GCE | 92.00 | 90.90 | 90.80 | 89.30 | 86.70 | 79.03 | 91.20 | 85.80 |
| | JoAPR | 88.32 | 87.85 | 87.00 | 87.03 | 84.55 | 80.15 | 82.79 | 69.02 |
| | NLPrompt | 91.73 | 91.13 | 90.77 | 89.93 | 88.30 | 86.70 | 91.17 | 89.27 |
| | NA-MVP | **92.07** | **92.10** | **91.60** | **91.30** | **90.07** | **89.37** | **91.47** | **89.53** |
| DTD | CoOp | 56.00 | 49.57 | 43.30 | 34.37 | 27.83 | 17.27 | 47.75 | 29.63 |
| | GCE | 61.00 | 59.83 | 56.80 | 50.73 | 43.60 | 33.67 | 57.57 | 43.97 |
| | JoAPR | 55.02 | 53.95 | 51.57 | 49.12 | 44.24 | 35.90 | 49.23 | 38.33 |
| | NLPrompt | 62.97 | 61.23 | 59.17 | 55.17 | 49.03 | 39.80 | 60.60 | 50.80 |
| | NA-MVP | **63.73** | **63.13** | **61.63** | **58.50** | **52.93** | **48.63** | **62.33** | **52.10** |
| Flowers102 | CoOp | 88.93 | 83.50 | 77.93 | 70.10 | 55.60 | 37.17 | 74.70 | 42.60 |
| | GCE | 88.80 | 88.33 | 86.73 | 84.07 | 78.37 | 70.37 | 86.37 | 69.93 |
| | JoAPR | 84.90 | 84.70 | 79.75 | 77.13 | 69.65 | 64.20 | 79.57 | 55.47 |
| | NLPrompt | 93.87 | 92.57 | **92.73** | 89.90 | 84.77 | **76.80** | **93.40** | **81.10** |
| | NA-MVP | **94.20** | **93.30** | 92.00 | **90.47** | **85.07** | 76.47 | 91.37 | 78.43 |
| OxfordPets | CoOp | 76.50 | 66.73 | 60.33 | 47.03 | 35.77 | 24.60 | 66.20 | 38.73 |
| | GCE | 85.63 | 84.60 | 83.67 | 79.23 | 71.40 | 53.17 | 83.03 | 68.07 |
| | JoAPR | 83.17 | 82.05 | 80.62 | 79.05 | 73.72 | 60.97 | 76.82 | 62.85 |
| | NLPrompt | 86.17 | 86.00 | 85.33 | 83.17 | 80.03 | 70.77 | 84.97 | 77.53 |
| | NA-MVP | **88.50** | **88.40** | **88.23** | **88.13** | **86.93** | **86.23** | **87.53** | **79.33** |
| UCF101 | CoOp | 69.03 | 63.40 | 58.23 | 49.73 | 40.83 | 26.30 | 58.07 | 34.43 |
| | GCE | 74.00 | 73.63 | 72.57 | 69.37 | 66.00 | 57.07 | 71.87 | 67.97 |
| | JoAPR | 70.80 | 69.22 | 68.15 | 64.80 | 61.82 | 57.52 | 63.98 | 49.67 |
| | NLPrompt | 74.83 | 73.40 | **72.83** | 70.33 | 68.10 | 60.53 | **73.58** | **65.97** |
| | NA-MVP | **75.33** | **74.03** | 72.30 | **70.93** | **68.43** | **63.93** | 73.40 | 65.40 |

**Effectiveness of Multi-view Prompts.** To validate the role of bi-directional multi-view prompt learning, we compare the following variants: Our baseline is CoOP with one prompt. *Negative Label*: Employs one clean and one noisy prompt per class, while explicitly assigning a negative label to each image. *Bi-directional Single Prompt*: Employs one clean and one noisy prompt per class, while treating non-target classes as implicit negatives. *Bi-directional Multi-view Prompts*: Our complete design using multiple clean and noisy prompts. As shown in Table 3 (a)-(d), employing both clean and noisy prompts already outperforms the single-prompt baseline, confirming that bi-directional supervision provides stronger robustness. However, the explicit negative-label strategy is less effective than our implicit negative design, as rigid counter-class assignments cannot generalize well under noisy conditions. Further introducing multi-view prompts enhances robustness, as it enriches semantic coverage and models intra-class diversity.

**Effectiveness of UOT for Patch-to-Prompt Alignment.** To assess the impact of UOT for local feature alignment, we compare it with different distribution alignment methods, *OT*: Enforces full mass matching between local features and prompts. *KL Divergence*: Minimizes divergence between prompt and image patch distributions. *UOT*: Introduces relaxed marginal constraints to focus on semantically relevant regions. As reported in Table 3 (e)-(g), UOT achieves the best performance in all noise ratios. KL divergence struggles to handle multi-modal distributions from diverse prompts, and OT suffers from over-constrained mass transport, often forcing irrelevant alignments. In contrast, UOT provides the necessary flexibility to softly align multiple local features to

Table 3: Ablation studies on DTD. (%)

| | Method/Noise Ratio | 25% | 50% | 75% | Avg |
|---|---|---|---|---|---|
| (a) | CoOp | 59.83 | 50.73 | 33.67 | 48.08 |
| (b) | (a)+ Negative label | 59.53 | 52.53 | 34.40 | 48.82 |
| (c) | (a)+ Bi-directional | 60.13 | 53.73 | 35.03 | 49.63 |
| (d) | (c)+ Multi-view | 62.73 | 55.13 | 37.63 | 51.83 |
| (e) | (d)+UOT | 62.50 | 57.70 | 42.33 | 54.18 |
| (f) | (d)+OT | 62.30 | 56.80 | 41.00 | 53.37 |
| (g) | (d)+KL Divergence | 62.27 | 56.43 | 39.10 | 52.60 |
| (h) | (e)+OT refinement | 59.60 | 54.77 | 45.77 | 53.38 |
| (i) | (h)+$\phi_{i,k}$ | **63.13** | **58.50** | **48.63** | **56.75** |

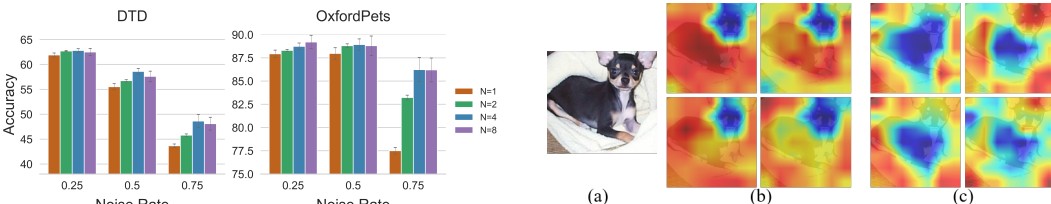

Figure 3: Test accuracy under varying label noise rates using different numbers of multi-view prompts $N \in \{1, 2, 4, 8\}$.

Figure 4: Visualization of bi-directional multi-view prompts. (a) The image; (b) The learned multi-view clean prompts; (c) The learned multi-view noisy prompts.

diverse prompt semantics. In addition, we provide further comparisons with other matching methods like NBNN (Boiman et al., 2008) in Appendix C.3.

**Effectiveness of Label Refinement.** We investigate the effectiveness of the proposed pseudo-label refinement strategy by comparing different variants of the global-level label correction mechanism: *Refine by OT (w/o $\phi_{i,k}$)*: this variant adopts a full label refinement strategy using OT, equivalent to using the label detection strategy used in NLPrompt and OT-Filter as noisy label identification strategy. *Refine by OT (w/$\phi_{i,k}$)*: partial refinement guided by $\phi_{i,k}$. As shown in Table 3, while full refinement improves performance under high noise levels, it can hurt accuracy when noise is low by mistakenly altering correct labels. In contrast, our selective refinement strategy (w/ $\phi_{i,k}$) consistently achieves better performance by focusing on likely noisy samples. Additional analysis of label refinement quality is provided in the Appendix C.2.

**Analysis of Multi-view Prompts.** To investigate the effect of the number of multi-view prompts, we evaluate NA-MVP under different numbers of multi-view prompts ($N \in \{1, 2, 4, 8\}$) across various noise rates on DTD and OxfordPets. As shown in Figure 3, performance improves as $N$ increases from 1 to 4, confirming the benefit of diverse semantic views. However, further increasing to $N = 8$ results in diminishing or even negative returns, suggesting that excessive prompts may cause redundancy. Thus, we adopt $N = 4$ as the default, balancing robustness and efficiency. We also analyze the effect of imbalanced numbers of clean and noisy prompts in Appendix C.1.

We also visualize the transport plan $T^*$ for four noisy and four clean prompts on the OxfordPets dataset in Figure 4. The heatmaps reveal that noisy and clean prompts attend to different object attributes, highlighting that $T^*$ captures discriminative patterns helpful for noisy label learning. Visualizations of failure cases are also provided in Appendix C.6.

We provide additional materials in the appendix, including method details, motivation analysis for noisy label identification, comparisons with other baselines, as well as hyper-parameter studies, generalization of NA-MVP, and computation cost evaluation.

## 5 CONCLUSION

In this paper, we presented NA-MVP, a framework for few-shot learning under noisy labels that integrates multi-view prompt learning with optimal transport. NA-MVP employs bi-directional multi-view prompts to capture both clean and noisy semantics, leverages Unbalanced Optimal Transport for fine-grained patch-to-prompt alignment, and applies classical Optimal Transport with a learnable threshold for selective label refinement. Experiments on diverse synthetic and real-world benchmarks show that NA-MVP achieves consistent improvements over strong baselines, especially in high-noise regimes. By coupling prompt learning with optimal transport, our approach offers a principled and effective solution for noise-robust few-shot learning. In future work, we plan to extend this framework to more challenging forms of noisy data, such as open-world supervision and complex real-world annotation noise, with potential applications in domains like medical imaging, video understanding, and cross-modal retrieval where robust few-shot learning is critical.

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

# A  METHOD DETAILS

---

**Algorithm 1** The training process of NA-MVP

---

1: **Input:** Noisy dataset $\mathcal{D}_{\text{noisy}}$, pretrained CLIP model $f$, $\psi(\cdot)$ and $\psi^n(\cdot)$, number of prompts $N$, entropy parameter $\epsilon$, total training epochs $T$, number of supervised epochs $T_{\text{sup}}$

2: **Output:** Optimized prompt parameter set $\Omega = \{\omega_m^c, \omega_m^n\}_{m=1}^{N}$ $t = 1, 2, \ldots, T$ each mini-batch $\mathcal{B}_t \subset \mathcal{D}_{\text{noisy}}$ $t > T_{\text{sup}}$

3: Identify noisy samples using threshold $\phi(p^n, p^c)$ and construct $\mathcal{D}_{\text{denoised}}$

4: Extract local feature map $F_i = \{f_l\}_{l=1}^{L} \in \mathbb{R}^{L \times d}$ for each $x_i \in \mathcal{B}_t$ using $f$

5: Generate prompt feature sets $G_k^c \in \mathbb{R}^{N \times d}$ and $G_k^n \in \mathbb{R}^{N \times d}$ for each class $k$

6: Compute cost matrices $C_k^c = 1 - F_i G_k^{c\top}$, $C_k^n = 1 - F_i G_k^{n\top}$ using cosine similarity

7: Solve UOT using Dykstra's algorithm:

$$T_k^{c*} = \text{diag}(\mu^{(t)}) \exp(-C_k^c/\epsilon) \text{diag}(\nu^{(t)}), \quad T_k^{n*} = \text{diag}(\mu^{(t)}) \exp(-C_k^n/\epsilon) \text{diag}(\nu^{(t)})$$

8: Compute UOT-based distances: $d_{\text{UOT}}^c(k) = \langle C_k^c, T_k^{c*} \rangle$, $d_{\text{UOT}}^n(k) = \langle C_k^n, T_k^{n*} \rangle$

9: Compute final prediction probabilities: $p(y = k \mid x_i) = (1 - p_{i,k}^n) \cdot p_{i,k}^c$ $t > T_{\text{sup}}$

10: Compute $\mathcal{L} = \mathcal{L}_{\text{gce}}$ using clean samples from $\mathcal{D}_{\text{denoised}}$

11: Compute $\mathcal{L} = \mathcal{L}_{\text{gce}} + \lambda_i \cdot \mathcal{L}_{\text{itbp}}$ using full dataset $\mathcal{D}_{\text{noisy}}$

12: Update prompt parameters $\Omega$ with loss $\mathcal{L}$ via SGD

13: $\Omega$

---

**Algorithm 2** Fast Implementation of Dykstra's Algorithm

---

**Input:** Cost matrix $C$, marginal vectors $\mu$, $\nu$, entropic regularization parameter $\epsilon$

1: **Initialize:** $Q \leftarrow \exp(-C/\epsilon)$, $\nu^{(0)} \leftarrow \mathbb{1}_\nu$, $\Delta_\nu \leftarrow \infty$, $\epsilon \leftarrow 10^{-3}$

2: Compute: $Q_\mu \leftarrow \dfrac{Q}{\text{diag}(\mu)\mathbb{1}_{|\mu| \times |\nu|}}$, $Q_\nu^\top \leftarrow \dfrac{Q^\top}{\text{diag}(\nu)\mathbb{1}_{|\mu| \times |\nu|}}$ $n = 1, 2, \ldots$

3: $\mu^{(n)} \leftarrow \min\left(\dfrac{\mathbb{1}_{|\mu|}}{Q_\mu \nu^{(n-1)}}, \mathbb{1}_{|\mu|}\right)$

4: $\nu^{(n)} \leftarrow \dfrac{\mathbb{1}_{|\nu|}}{Q_\nu^\top \mu^{(n)}}$

5: $\Delta_\nu \leftarrow \left\| \nu^{(n)} - \nu^{(n-1)} \right\|$ $\Delta_\nu < \epsilon$

6: **break**

7: $T^* = \text{diag}(\mu^{(n)}) Q \, \text{diag}(\nu^{(n)})$

---

**Training Process**  We detail the training process of NA-MVP in Algorithm 1, which includes a supervised phase and a denoising phase.

**Optimal Transport.**  OT is a powerful framework for mapping one probability distribution to another while minimizing the associated transportation cost. Given two distributions $\boldsymbol{\mu} \in \mathbb{R}_+^m$ and $\boldsymbol{\nu} \in \mathbb{R}_+^n$, and a cost matrix $\boldsymbol{C} \in \mathbb{R}^{m \times n}$, the OT problem aims to find the optimal transport plan $\boldsymbol{T}$ that minimizes the following objective:

$$d_{\text{OT}}(\mu, \nu) = \min_{\boldsymbol{T} \in \Pi(\boldsymbol{\mu}, \boldsymbol{\nu})} \langle \boldsymbol{C}, \boldsymbol{T} \rangle \tag{21}$$

$$\Pi(\mu, \nu) = \left\{ \mathbf{T} \in \mathbb{R}_+^{m \times n} \mid \mathbf{T}\mathbb{1}_n = \mu, \ \mathbf{T}^\top \mathbb{1}_m = \nu \right\} \tag{22}$$

where $\langle \cdot, \cdot \rangle$ represents the Frobenius dot-product, and $\mathbb{1}_m$, $\mathbb{1}_n$ denote the vectors of ones of length $m$ and $n$, respectively. Since solving OT exactly is computationally expensive, the entropy-regularized version is often used:

$$d_{\text{OT}}(\mu, \nu) = \min_{\boldsymbol{T} \in \Pi(\boldsymbol{\mu}, \boldsymbol{\nu})} \langle \boldsymbol{C}, \boldsymbol{T} \rangle + \epsilon \langle \boldsymbol{T}, \log \boldsymbol{T} \rangle \tag{23}$$

where $\epsilon > 0$ controls the strength of regularization. The added entropy term $\langle \boldsymbol{T}, \log \boldsymbol{T} \rangle$ promotes smoother transport plans and allows for efficient optimization via the Sinkhorn algorithm (Distances,

2013). The optimization process can be completed in a few iterations, with the solution $T^*$ being computed as:

$$T^* = \text{diag}(\mu^{(t)}) \exp(-\boldsymbol{C}/\epsilon) \text{diag}(\nu^{(t)}), \tag{24}$$

where $t$ denotes the iteration number and in each iteration, the marginal distributions $\mu^{(t)} = \mu/\left(\exp(-\boldsymbol{C}/\epsilon)\nu^{(t-1)}\right)$ and $\nu^{(t)} = \nu/\left(\exp(-\boldsymbol{C}/\epsilon)\mu^{(t)}\right)$.

**Fast Implementation of Dykstra's Algorithm**  To efficiently solve the entropically regularized UOT problem defined in Eq. 3, we adopt a fast matrix-scaling variant of Dykstra's algorithm. The full procedure is detailed in Algorithm 2.

**Generalized Cross-Entropy Loss.**  To support the training objectives described in Eq. (26)–(27), we provide a detailed explanation of the Generalized Cross-Entropy (GCE) loss (Zhang & Sabuncu, 2018), denoted as $\mathcal{L}_{\text{gce}}$. This loss serves as the primary supervision signal for training, especially in the presence of label noise.

The GCE loss is a noise-robust surrogate to the standard cross-entropy (CE) and mean absolute error (MAE) losses. Given a training sample $(x, y)$ where $y \in \{1, 2, \ldots, C\}$ is the ground-truth label and $p = f(x) \in \Delta^{C-1}$ is the softmax output over $C$ classes, the GCE loss is defined as:

$$\mathcal{L}_{\text{gce}}(x, y) = \frac{1 - p_y^q}{q}, \qquad 0 < q \leq 1, \tag{25}$$

where $p_y$ denotes the predicted probability for class $y$, and $q$ is a tunable hyper-parameter that governs the degree of robustness. Following prior works, we fix $q = 0.5$ throughout all experiments, which offers a good trade-off between noise robustness and optimization stability.

## B  EXPERIMENTAL DETAILS

### B.1  DATASET DETAILS

We selected six representative visual classification datasets as benchmarks. The detailed statistics of each dataset are shown in Table 4, including the original task, the number of classes, and the sizes of training and test samples.

Table 4: The detailed statistics of datasets used in experiments.

| Dataset | Task | Classes | Training Size | Testing Size |
|---|---|---|---|---|
| Caltech101 (Fei-Fei et al., 2004) | Object recognition | 100 | 4,128 | 2,465 |
| DTD (Cimpoi et al., 2014) | Texture recognition | 47 | 2,820 | 1,692 |
| Flowers102 (Nilsback & Zisserman, 2008) | Fine-grained flowers recognition | 102 | 4,093 | 2,463 |
| OxfordPets (Parkhi et al., 2012) | Fine-grained pets recognition | 37 | 2,944 | 3,669 |
| UCF101 (Peng et al., 2018) | Video action recognition | 101 | 7,639 | 3,783 |
| Food101N (Lee et al., 2018) | Fine-grained food recognition | 101 | 310,009 | 30,300 |

### B.2  IMPLEMENTATION DETAILS

All input images are resized to $224 \times 224$ and divided into $14 \times 14$ patches of dimension 768. For the Unbalanced OT problem in Eq. 6, we set the entropic regularization weight to $\epsilon = 0.1$ and the marginal relaxation parameter to $\theta = 0.9$. The loss balancing coefficient $\lambda_{\text{i}}$ is set to 0.1. The maximum number of iterations in Algorithm 2 is set to 100, with early stopping applied when $\Delta_\nu < 0.01$. We use 16 shared context tokens appended to the class token, each of dimension 512. Prompts are randomly initialized and inserted at the "end" token position. Batch sizes are set to 32 for training and 100 for testing. The total number of training epochs is 50, with 20 for the supervised phase and 30 for semi-supervised refinement. Warm-up is set to 1 epoch for all datasets, except for the Flowers dataset, which adopts a 20-epoch warm-up. All experiments are conducted on a single NVIDIA GeForce RTX 4090 GPU.

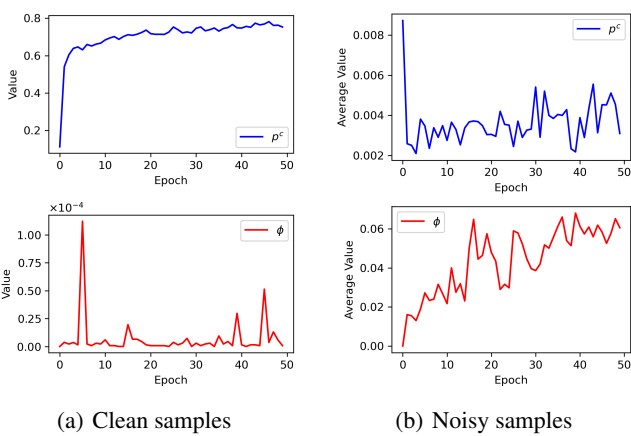

(a) Clean samples          (b) Noisy samples

Figure 5: Behavioral differences of $p^c$ and $\phi$ on clean vs. noisy samples in Caltech101.

Table 5: Experimental results under imbalanced numbers of prompts on DTD and OxfordPets.

| Datasets | DTD | | | | OxfordPets | | | |
|---|---|---|---|---|---|---|---|---|
| Noise rate | 25% | 50% | 75% | Avg. | 25% | 50% | 75% | Avg. |
| N=1 | 62.36 | 57.33 | 47.24 | 55.64 | 88.27 | 87.23 | 85.26 | 86.92 |
| N=2 | 62.86 | 57.17 | 47.40 | 55.81 | 88.20 | 87.59 | 85.50 | 87.10 |
| N=4 | **63.13** | **58.50** | **48.63** | **56.75** | 88.40 | **88.13** | **86.23** | **87.59** |
| N=8 | 62.76 | 57.07 | 47.37 | 55.73 | **88.57** | 87.63 | 85.73 | 87.31 |

### B.3 Noisy label identification motivation

To validate the intuition behind our noisy label identification strategy, we conduct a case study on the Caltech101 dataset. Samples are grouped into clean and noisy subsets based on their ground-truth annotations. We visualize the epoch-wise average values of the clean prompt confidence $p^c$ and the adaptive threshold $\phi$ for both groups.

As shown in Figure 5, clean samples exhibit consistently high values of $p^c$, while their corresponding $\phi$ remains close to zero. In contrast, noisy samples demonstrate the opposite pattern: $\phi$ is significantly larger than $p^c$, especially in the early training phase. These trends confirm that our bi-directional alignment framework provides a meaningful signal to distinguish between clean and noisy labels. Moreover, the observed dynamics further justify the design of our selective noisy label refinement strategy introduced in Section 3.2, which integrates soft thresholding with OT-based pseudo-label correction.

## C Additional Experimental results

### C.1 Analysis of the Number of Multi-view Prompts

To investigate the effect of imbalanced numbers of clean and noisy prompts, we conduct experiments by fixing the number of clean prompts to 4 while varying the number of noisy prompts (N = 1, 2, 4, 8). The evaluation is performed under different noise rates on the DTD and OxfordPets datasets. The results clearly show that the best performance is achieved when the number of noisy prompts equals the number of clean prompts (i.e., N=4). This trend is particularly evident under higher noise rates, indicating that a balanced prompt configuration enhances model robustness to noise.

### C.2 Effectiveness of Selective Refinement with $\phi_{i,k}$

To further analyze the effectiveness of our proposed selective label refinement guided by $\phi_{i,k}$, we compare the evolution of noisy label ratio over training epochs under different settings. As shown

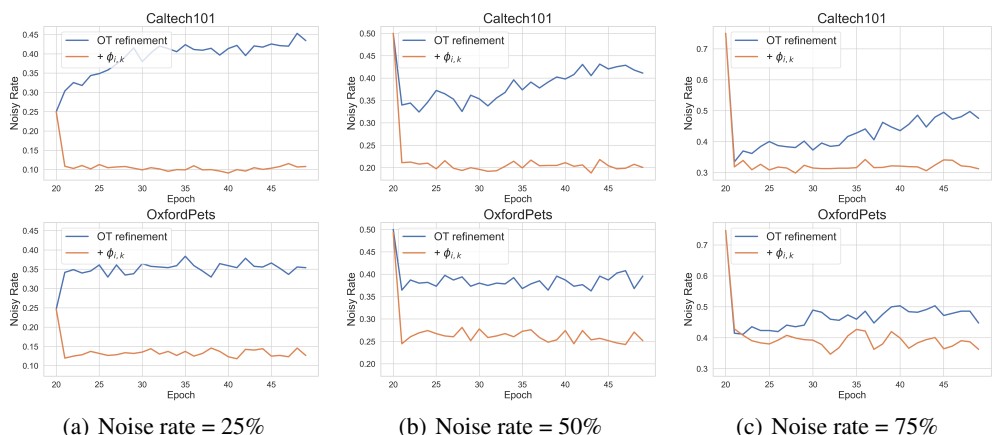

(a) Noise rate = 25%            (b) Noise rate = 50%            (c) Noise rate = 75%

Figure 6: Comparison of OT refinement and selective refinement with $\phi_{i,k}$ under different noise rates on Caltech101 and OxfordPets datasets.

Table 6: The accuracy comparison on OxfordPets Dataset.

| Method/Noise rate | 12.5% | 25% | 37.5% | 50% | 62.5% | 75% |
|---|---|---|---|---|---|---|
| DEFT (Wei et al., 2024) | **88.83%** | 88.23% | 88.10% | 86.73% | 84.10% | 75.87% |
| NA-MVP | 88.50% | **88.40%** | **88.23%** | **88.13%** | **86.93%** | **86.23%** |

in Figure 6, we visualize the noisy rate curves during training on Caltech101 and OxfordPets under three synthetic noise levels: 25%, 50%, and 75%. We compare the baseline OT refinement strategy with our selective refinement method guided by $\phi_{i,k}$. Across all datasets and noise settings, we observe that the noisy rate decreases more significantly and remains consistently lower when using $\phi_{i,k}$-guided refinement. This indicates that our method effectively filters out potentially clean samples from aggressive relabeling, preventing overcorrection and improving the quality of pseudo-labels. These findings further demonstrate that $\phi_{i,k}$ improves label correction reliability by enabling a more conservative and noise-aware refinement process.

Additionally, we conducted further experiments on the OxfordPets dataset to compare our method with DEFT (Wei et al., 2024). DEFT selects clean samples using the rule: $p_{ik}^{\text{clean}} > 0.5$. The results, summarized in Table 6, show that NA-MVP consistently outperforms DEFT across all noise levels, particularly under higher noise ratios. These results reinforce the effectiveness of our method in handling noisy data, especially in scenarios with high noise levels, where our adaptive approach offers a significant improvement over the baseline methods.

## C.3 COMPARISON WITH OTHER METHODS

**Comparison with PLOT**    PLOT (Chen et al., 2022) utilizes OT to align image and prompt features for few-shot classification and inspired us to identify noisy labels from multi-view. While PLOT has shown great promise, our method, NA-MVP, improves upon this by integrating multi-view prompt learning and a UOT-based denoising strategy, particularly effective in high-noise conditions. To validate NA-MVP, we compared it with PLOT under various noise levels. As shown in Table 7, NA-MVP consistently outperforms PLOT in all datasets and noise settings, with a particularly significant advantage under higher noise conditions. These results emphasize the robustness of our framework in handling noisy labels, supporting the effectiveness of our approach in noisy regimes.

**Comparison with unsupervised methods**    Although few-shot learning generally outperforms unsupervised methods, we recognize the importance of empirically evaluating whether learning from noisy few-shot labels can yield meaningful improvements. In this regard, we compared NA-MVP with two recent unsupervised methods: MetaPrompt (Mirza et al., 2024) and LaFTer (Mirza et al., 2023). As shown in Table 8, NA-MVP outperforms both MetaPrompt and LaFTer across most

Table 7: The accuracy comparison across datasets and noise levels.

| Dataset | Method | Noise rate: Sym | | | | | | Noise rate: Asym | |
|---|---|---|---|---|---|---|---|---|---|
| | | 0.125 | 0.25 | 0.375 | 0.5 | 0.625 | 0.75 | 0.25 | 0.5 |
| Caltech101 | PLOT | 90.03 | 88.10 | 85.13 | 84.10 | 75.90 | 62.70 | 81.17 | 50.80 |
| | NA-MVP | **92.07** | **92.10** | **91.60** | **91.30** | **90.07** | **89.37** | **91.47** | **89.53** |
| DTD | PLOT | 60.27 | 56.87 | 50.77 | 44.67 | 36.53 | 23.57 | 52.80 | 32.03 |
| | NA-MVP | **63.73** | **63.13** | **61.63** | **58.50** | **52.93** | **48.63** | **62.33** | **52.10** |
| Flowers102 | PLOT | 91.63 | 89.00 | 84.67 | 77.10 | 66.80 | 47.57 | 76.20 | 40.60 |
| | NA-MVP | **94.20** | **93.30** | **92.00** | **90.47** | **85.07** | **76.47** | **91.37** | **78.43** |
| OxfordPets | PLOT | 84.57 | 79.43 | 74.70 | 64.60 | 52.10 | 41.50 | 73.87 | 44.47 |
| | NA-MVP | **88.50** | **88.40** | **88.23** | **88.13** | **86.93** | **86.23** | **87.53** | **79.33** |
| UCF101 | PLOT | 73.30 | 69.37 | 65.27 | 59.13 | 51.50 | 40.93 | 61.33 | 36.43 |
| | NA-MVP | **75.33** | **74.03** | **72.30** | **70.93** | **68.43** | **63.93** | **73.40** | **65.40** |

Table 8: Performance comparison with unsupervised medthods.

| Method | OxfordPets | DTD | UCF101 | Flowers102 |
|---|---|---|---|---|
| MetaPrompt (Mirza et al., 2024) | 88.10 | 50.80 | 67.90 | 73.90 |
| LaFTer (Mirza et al., 2023) | 82.70 | 46.10 | 68.20 | 71.00 |
| NA-MVP (25% noise) | **88.40** | **63.10** | **74.00** | **93.30** |
| NA-MVP (50% noise) | **88.13** | **58.50** | **70.93** | **90.47** |
| NA-MVP (75% noise) | 86.23 | 48.63 | 63.93 | **76.47** |

datasets. Even at higher noise levels (50% and 75%), our method remains competitive or superior. These results demonstrate that learning from a small amount of noisy supervision, when appropriately modeled, can be more effective than training with no labeled data. This underscores the practical value of noisy few-shot learning in real-world low-resource scenarios.

**Comparison with NBNN**  We also compared NA-MVP with NBNN (Boiman et al., 2008), a widely used set-to-set matching method with cosine similarity as a distance metric. Our experiments on the Caltech101 and OxfordPets datasets show that NA-MVP consistently outperforms NBNN across different noise levels, particularly under higher noise conditions. The results in Table 9 indicate that UOT-based matching in NA-MVP provides a more robust and adaptive alignment between prompts and image features, especially when class distributions are corrupted or ambiguous.

Table 9: Performance comparison on Caltech101 and OxfordPets.

| Dataset | Method | 12.5% | 25.0% | 37.5% | 50.0% | 62.5% | 75.0% |
|---|---|---|---|---|---|---|---|
| Caltech101 | NBNN | 88.87 | 88.57 | 88.43 | 87.20 | 84.17 | 85.67 |
| | NA-MVP | **92.07** | **92.10** | **91.60** | **91.30** | **90.07** | **89.37** |
| OxfordPets | NBNN | 87.47 | 86.00 | 85.53 | 83.07 | 82.50 | 80.67 |
| | NA-MVP | **88.50** | **88.40** | **88.23** | **88.13** | **86.93** | **86.23** |

## C.4   PARAMETER STUDY

**Parameter Study of the Auxiliary Loss Weight** $\lambda_i$   We study the effectiveness of the auxiliary loss weight $\lambda_i$, which controls the contribution of the ITBP loss during supervised training. As shown in Table 10, we evaluate $\lambda_i \in \{0.01, 0.05, 0.1, 0.5\}$ across DTD, OxfordPets, and UCF101 under different noise levels. In particular, $\lambda_i = 0.1$ consistently delivers the best or competitive results under different conditions. Therefore, we adopt $\lambda_i = 0.1$ as the default setting in all experiments.

Table 10: Impact of the loss balancing coefficient $\lambda_i$ under different noise rates.

| Datasets | DTD | | | OxfordPets | | | UCF101 | | |
|---|---|---|---|---|---|---|---|---|---|
| Noise rate | 25% | 50% | 75% | 25% | 50% | 75% | 25% | 50% | 75% |
| $\lambda_i$=0.01 | 62.36 | 57.96 | 48.09 | 88.37 | 87.93 | 85.43 | 73.53 | 70.57 | 61.87 |
| $\lambda_i$=0.05 | 62.03 | 58.10 | 48.39 | **88.53** | **88.37** | 85.90 | 73.60 | **71.09** | 62.90 |
| $\lambda_i$=0.1 | **63.13** | **58.50** | **48.63** | 88.40 | 88.13 | **86.23** | **74.03** | 70.93 | **63.93** |
| $\lambda_i$=0.5 | 62.96 | 57.33 | 47.19 | 88.33 | 87.50 | 85.33 | 73.63 | 69.13 | 60.74 |

Table 11: Performance under different noise rates and $\theta$ values.

| Noise rate/$\theta$ | 0.5 | 0.6 | 0.7 | 0.8 | 0.9 | 1.0 |
|---|---|---|---|---|---|---|
| 25.00% | 61.87 | 62.83 | 63.30 | **63.80** | 63.13 | 63.37 |
| 50.00% | 57.57 | 58.73 | 58.07 | 58.67 | **58.80** | 58.30 |
| 75.00% | 47.43 | 48.30 | 47.67 | 47.13 | **48.63** | 46.30 |

**Parameter Study of the parameter $\theta$ in UOT**  To investigate the effect of the parameter $\theta$ in unbalanced OT, which regulates the mapping size of prompts on the feature map, we conducted additional experiments on the DTD dataset under varying noise rates and $\theta$ values ranging from 0.5 to 1.0. As shown in Table 11, we observe that performance varies with $\theta$, and optimal results are typically achieved when $\theta$ is within the range of 0.8–0.9 across different noise levels. This indicates that optimal alignment between multi-view prompts and the feature map is achieved when approximately 80%–90% of patch tokens are involved in the prompt interaction. Consequently, we adopt $\theta = 0.9$ as the default setting in all our main experiments to ensure a good balance between sufficient prompt supervision and robustness under noise.

## C.5   EXPERIMENTS ON WATERBIRDS DATASET

To further assess robustness, we conducted experiments on the Waterbirds dataset (Sagawa et al.) under multiple levels of label noise. The Waterbirds dataset is a common benchmark for studying spurious correlations, as its backgrounds (water or land) are strongly associated with class labels and often occupy a large portion of the image. This makes it suitable for evaluating sensitivity to background–foreground imbalance or noisy supervision. As shown in Table 12, accuracy decreases as the noise level increases, yet NA-MVP consistently outperforms NLPrompt across all settings. These results suggest that NA-MVP is relatively robust to mislabeled data and performs well on datasets with both small and large background regions, benefiting from its unbalanced optimal transport formulation, which enables the model to downweight irrelevant or misleading signals.

Table 12: Accuracy comparison on the Waterbirds dataset.

| Method / Noise rate | 12.5% | 25% | 37.5% | 50% | 62.5% | 75% |
|---|---|---|---|---|---|---|
| NLPrompt | 74.23 | 72.47 | 71.50 | 68.43 | 64.80 | 58.27 |
| NA-MVP | **75.27** | **74.40** | **72.07** | **69.27** | **65.47** | **59.23** |

## C.6   ANALYSIS OF FAILURE CASES

To better understand the limitations of our method, we further visualize failure cases on the Oxford-Pets dataset, as shown in Figure 7. The figure illustrates attention maps of several learned clean and noisy prompts. We observe that in these cases the model fails to clearly distinguish between clean- and noise-oriented prompts. For example, the bottom-right clean prompt in (b2) and the bottom-right noisy prompt in (b3) exhibit highly similar activation patterns, indicating that the intended separation between clean and noisy supervision is not well preserved. Moreover, the heatmaps reveal that many prompts predominantly focus on background regions rather than the object of interest. This

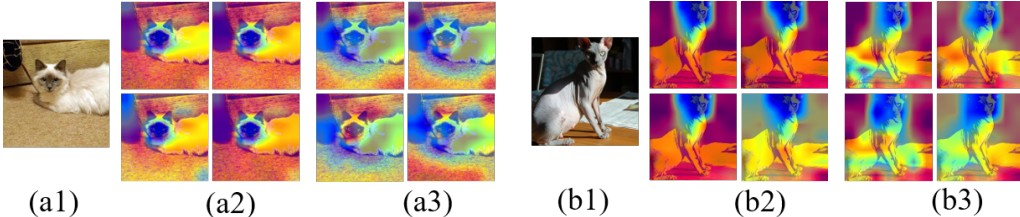

$$\text{(a1)} \qquad \text{(a2)} \qquad \text{(a3)} \qquad \text{(b1)} \qquad \text{(b2)} \qquad \text{(b3)}$$

Figure 7: Visualization of failure cases on OxfordPets. (a1) & (b1): The image; (a2) & (b2): The learned multi-view clean prompts; (a3) & (b3): The learned multi-view noisy prompts.

Table 13: The generalization of NA-MVP.

| Method/Noise rate | 12.5% | 25% | 37.5% | 50% | 62.5% | 75% |
|---|---|---|---|---|---|---|
| VPT | 57.50 | 55.37 | 50.70 | 45.03 | 36.93 | 25.27 |
| **VPT+Ours** | **68.43** | **66.93** | **66.10** | **63.57** | **60.00** | **53.80** |
| MaPLe | 63.27 | 55.00 | 49.07 | 40.20 | 32.67 | 19.93 |
| **MaPLe+Ours** | **69.70** | **67.50** | **65.37** | **62.83** | **56.47** | **45.83** |

misalignment reduces the effectiveness of prompt-feature alignment, leading to incorrect label predictions. Such cases highlight that the current design may overfit to spurious background cues when discriminative foreground signals are weak or ambiguous, suggesting the need for more flexible and accurate prompt learning mechanisms in future work.

## C.7 GENERALIZATION OF NA-MVP

To further demonstrate the generalization capability of our NA-MVP framework, we apply it to two representative prompt-tuning methods beyond CoOp: VPT (Jia et al., 2022) and MaPLe (Khattak et al., 2023a). As shown in Table 13, NA-MVP consistently improves their performance on the DTD dataset under various symmetric noise levels, demonstrating its strong generalization.

## C.8 COMPUTATION COST EVALUATION

We compare the inference time of NA-MVP with the baseline method CoOp and NLPrompt on OxfordPets. As reported in Table 14, NA-MVP achieves the fastest inference time of 5.778 seconds, which is 25.2% faster than CoOp and 79.6% faster than NLPrompt. These results demonstrate that NA-MVP maintains competitive computational efficiency despite its multi-view prompt design.

While NA-MVP requires more training time, the increase remains within a practical range. Given the consistent performance improvements across varying noise levels, the method offers a reasonable trade-off between computational cost and robustness, making it suitable for real-world applications where label noise is prevalent.

Table 14: The time cost comparison.

| Settings | CoOp | NLPrompt | NA-MVP |
|---|---|---|---|
| Training Time (s) | 1.875 | 4.394 | 6.285 |
| Inference Time (s) | 7.719 | 28.276 | 5.778 |

## D LLM USAGE

Standard editing tools, including large language models, were used occasionally to polish the presentation of the manuscript (e.g., grammar, spelling, and word choice). Their use was limited to light language refinement, and they did not contribute to the conceptual development, methodology, experiments, or analysis of this work.

