# OpenReview forum: "Noise-Aware Few-Shot Learning through Bi-directional Multi-view Prompt Alignment"
_ICLR.cc/2026/Conference — ICLR 2026 Conference Withdrawn Submission_

### Official Review · Reviewer_4JvG · 2025-10-30

**Soundness:** 2
**Presentation:** 1
**Contribution:** 3
**Rating:** 2
**Confidence:** 4

**Summary:**

This paper addresses few-shot prompt learning with noisy labels. The method learns two sets of prompts per class: clean-oriented prompts aligned with semantically meaningful local features, and noise-aware prompts aligned with ambiguous or irrelevant features, using unbalanced Optimal Transport (UOT) for local alignment. Images whose representations align more strongly with noise-aware prompts than with clean prompts are flagged as potentially mislabeled through an adaptive threshold. Then, classical OT is applied once over the entire training set to obtain a global transport plan between all image features and all class prompt features. Only for the samples flagged as noisy, the original labels are replaced with the pseudo-labels inferred from this transport plan, while the labels of samples considered clean are kept unchanged.

**Strengths:**

- **Strong empirical performance**

    The method demonstrates consistent improvements over relevant baselines across multiple noisy few-shot benchmarks, indicating practical effectiveness.

- **Well-motivated focus on noisy few-shot adaptation**

    Addressing label noise in few-shot prompt tuning is a meaningful and realistic challenge, and the paper proposes a systematic pipeline that combines local alignment signals with selective label refinement.

- **Integration of OT mechanisms into prompt learning**

    The use of unbalanced optimal transport for local feature–prompt alignment, combined with global OT-based pseudo-label refinement, offers an interesting and principled perspective on leveraging OT in the context of noisy prompting.

**Weaknesses:**

- **Clarity of the writing**

    The clarity and organization of the manuscript could be improved. Some methodological components are difficult to follow, and several elements would benefit from more explicit definitions and explanations. For example, certain similarity terms are used before being formally introduced, and the Image-Text Bi-Directional Prompt loss is only described conceptually without a precise mathematical formulation. In addition, the description of some baselines and ablations in the experimental section lacks sufficient detail, which makes it challenging to fully understand the comparison setup. Some of the technical derivations related to the unbalanced OT formulation and the classical OT refinement could be moved to the supplementary material to improve the readability of the main text. The freed space could then be used to clarify other key components of the method, in particular the generalized cross-entropy loss and the ITBP loss, which currently lack sufficient detail and explicit formulation.

- **Positioning with respect to prior work**

    While the method targets a relevant problem and achieves strong empirical results, its positioning relative to prior work could be clarified further. Optimal transport for prompt–region alignment has been explored in PLOT [1], and OT-based label refinement for noisy supervision in NLPrompt [2]. Moreover, the idea of introducing negative or noise-aware prompts has been investigated in the OOD detection literature [3,4,5]. The present work applies these ideas in the context of noisy few-shot learning and introduces improvements such as unbalanced OT for local alignment and selective relabeling. These extensions are meaningful, but may be viewed as incremental unless their advantages over prior approaches are articulated more clearly. Clarifying the conceptual contribution and differentiating more explicitly from prior art would strengthen the paper.

[1] Chen et al, "PLOT: Prompt Learning With Optimal Transport For Vision-language Models", ICLR 2023

[2] Pan et al, "NLPrompt: Noise-Label Prompt Learning for Vision-Language Models", CVPR 2025

[3] Bai et al., ID‐like Prompt Learning for Few‐Shot Out‐of‐Distribution Detection, CVPR 2024

[4] Li et al., Learning Transferable Negative Prompts for Out‐of‐Distribution Detection, CVPR 2024

[5] Nie et al., Out-of-Distribution Detection with Negative Prompts, ICLR 2024

**Questions:**

1. Why are two separate text encoders used for clean and noise-aware prompts? Are they trained independently or share parameters? What advantage does this bring compared to a single encoder?
2. Could you explicitly provide the mathematical definition of the ITBP loss and specify how clean and noisy prompts participate as positive and negative pairs?
3. In Section “Effectiveness of Multi-view Prompts,” what does “treating non-target classes as implicit negatives” mean concretely in the contrastive setup? In the same section, “Negative Label” is described as assigning a negative label to each image. How is this implemented?
4. In the “Bi-directional Multi-view Prompts” configuration, UOT is not applied. How is the alignment between image patches and prompts performed in this case?
5. Why compare $\phi_{i,k}$ (a softmax probability) directly to $s^c_{i,k}$ (a similarity score) in Eq 10 and Eq 14 instead of comparing probabilities on the same scale (e.g.,$ \phi_{i,k} > 0.5$)? How sensitive are results to the temperature $\tau$?
6. In Figure 4, several clean prompts appear not to attend to salient regions (e.g., the dog’s head), while some noise prompts do. How do the authors interpret this behavior and its relation to the claimed separation between clean and noisy cues? Is this an example of a image with a noisy label?
7.  The use of OT for prompt-region alignment was introduced in PLOT [1], and OT-based label refinement in NLPrompt [2]. Could the authors clarify how their contributions go beyond these works? My understanding is that the main novelties lie in (i) the use of unbalanced OT for local alignment, (ii) selective label correction applied only to samples detected as noisy, and (iii) the integration of noise-aware prompts.
8. Can the authors provide results on clean data only and compare to state-of-the-art prompt learning methods such as PromptSRC [3], ProDA [4] or GalLoP [5]?
9. Is it possible that the same region of the image is assigned to a positive prompt and a negative prompt simultaneously?

[1] Chen et al, "PLOT: Prompt Learning With Optimal Transport For Vision-language Models", ICLR 2023

[2] Pan et al, "NLPrompt: Noise-Label Prompt Learning for Vision-Language Models", CVPR 2025

[3]  Khattak et al, "Self-regulating Prompts: Foundational Model Adaptation without Forgetting", ICCV 2023

[4] Lu et al, "Prompt Distribution Learning", CVPR 2022

[5] Lafon et al, "GalLoP: Learning global and local prompt for vision-language models", ECCV 2024

---

### Official Review · Reviewer_SPrg · 2025-10-31

**Soundness:** 1
**Presentation:** 2
**Contribution:** 1
**Rating:** 2
**Confidence:** 4

**Summary:**

The paper mainly focuses on addressing the issues caused by noisy labels, and thereby proposes three limitations and a framework based on bi-directional Multi-View Prompt alignment to enable Vision-language models to generalize well in downstream few-shot learning.

**Strengths:**

1. The works listed in the "Related Work" section are quite recent.
2. The compared SOTAs are quite new.

**Weaknesses:**

1. The overall logic falls somewhat short. The three sub-issues proposed in this work have weak connections to the key problem of noisy labels that needs to be addressed.
2. Moreover, there is no solid evidence demonstrating that the authors' method effectively resolves these three sub-issues.
3. The writing is poor and makes it hard to understand. The implementation details in the method section are not clearly explained.

For the specific drawbacks, please refer to the "Questions" section below.

**Questions:**

1. The paper mentions three issues with current prompt learning methods when dealing with noisy labels. The first problem pointed out by the authors is "a single-view alignment that fails to capture diverse and fine-grained cues." However, the inability to capture fine-grained features is not exclusively caused by noisy labels—it is a rather common issue. What is the connection between this problem and the noisy labels addressed by the authors? Additionally, how does the Multi-view Prompt alignment method proposed in the methodology section relate to noisy labels, and why can it be used to solve the problem of noisy labels?
2. Similarly, in Figure 1, the authors propose three corresponding limitations of existing methods. However, the authors do not clearly elaborate on these issues, nor do they explain or prove whether noisy labels are solely responsible for these problems, and do not exist in clean labels. Furthermore, the subsequent experiments fail to demonstrate that the proposed method effectively addresses these three issues in a corresponding manner. Thus, it is demonstrated whether these problems no longer exist.
3. In the methodology section, what is UOT? Why can UOT be used to address overfitting to noise? How does partial alignment help in solving overfitting to noise? There is a lack of effective argumentation here.
4. Since this is a setting for few-shot learning, is it feasible to train using two separate text encoders with such a limited amount of data?
5. Table 1 compares too few existing methods aimed at the noisy label problem, which is insufficient to support the effectiveness of the authors' approach. It would be better to include more recent methods for comparison. Furthermore, it should be clarified under what kind of few-shot settings. Additional settings, such as 1-shot or 5-shot, could be included.

---

### Official Review · Reviewer_CsJc · 2025-11-01

**Soundness:** 2
**Presentation:** 3
**Contribution:** 2
**Rating:** 4
**Confidence:** 2

**Summary:**

This paper introduces NA-MVP, a noise-aware few-shot learning framework that integrates bi-directional multi-view prompt alignment with optimal transport. The method jointly learns clean-oriented and noise-aware prompts to separate reliable signals from corrupted ones and applies selective label refinement with a learnable threshold for robust training. Experiments on synthetic and real-world datasets demonstrate  performance gains over baselines, validating the effectiveness of NA-MVP under noisy supervision.

**Strengths:**

- The paper presents a well-motivated framework that addresses three challenges of noisy few-shot learning.
- Comprehensive analytical experiments on multiple datasets support the proposed contributions.

**Weaknesses:**

- The proposed method does not yield consistent performance improvements over the baseline NLPrompt in table 1, especially when facing the asymmetric noises.
- In table 2, as the number of shots increases, the performance gap between the proposed NA-MVP and the baseline NLPrompt becomes smaller. Does this suggest that the advantages of noise-aware alignment diminish when more samples are available? Not ensure whether the method could scale when data volume grows.

**Questions:**

- How does the method guarantee that multiple prompts capture different aspects of the visual categories?

---

### Official Review · Reviewer_ZBbe · 2025-11-09

**Soundness:** 2
**Presentation:** 2
**Contribution:** 2
**Rating:** 4
**Confidence:** 3

**Summary:**

This paper proposes a new approach for prompt learning based on Optimal Transport (OT). This so-called NA-MVP approach aims at improving contrastive Vision Language Models (VLM) such as CLIP's robustness toward noisy labels. To do so, three main mechanisms are introduced: a bidirectional local visual and textual features alignment based on unbalanced OT, noisy label identification through an adaptive threshold on global visual-textual similarity, and OT-based label refinement. Experiments are presented on standard prompt learning benchmarks.

**Strengths:**

The method is clear with quality illustrations.
The method shows consistent gains against the selected baselines.
Leveraging prompt learning to tackle noisy labels is an interesting proposition that has yet not be extensively studied.

**Weaknesses:**

Several recent works tackle OT for prompt learning. FedOTP (CVPR24) proposes using unbalanced OT for global and local prompt cooperation, while PatchCT (ICCV23) introduces conditional transport for aligning visual and textual tokens at scale. Other recent work, such as GalLop (ECCV24) or LoCoOp (Neurips23), also proposes robust local/global mechanisms to enhance CLIP's robustness. Due to the proximity of these works to the proposed approach, a better positioning and a broader comparison with the current state-of-the-art would be required.

From Table 3, it appears that most of the gains are attributed to the multi-view bi-bi-directional prompt, which was introduced previously in CLIPN. Furthermore, the method introduces various losses to align and filter prompts, but OT refinement appears to be actually detrimental. Finally, among the paper's contributions, most of the gains appear to be provided by the thresholding from eq. (9 - 10).

The method's stability toward parameter $\lambda_i$ is difficult to assess as it is only evaluated for four different values. At the very least $\lambda_i = 0$ should be reported.

**Questions:**

I wonder what is the total number of hyperparameters of the method.

What is the impact of the parameter $\tau$ on the method?

Prompt learning is often viewed as a cheap and easy way to adapt a VLM. On the contrary, here, the proposed method here is relatively complex. What is the inference time or FLOPS for the approach compared to the other prompt learning baselines?

---

### Note · Authors · 2025-11-12

I have read and agree with the venue's withdrawal policy on behalf of myself and my co-authors.